# A sharp transition in quantum chaos and thermodynamics of mass deformed SYK model

T. Nosaka*[1,2],

**1** INFN Sezione di Trieste, 34127 Trieste, Italy
**2** International School for Advanced Studies (SISSA), 34136 Trieste, Italy
* nosaka@yukawa.kyoto-u.ac.jp

October 23, 2020

## Abstract

We review our recent work [1] where we studied the chaotic property of the two coupled Sachdev-Ye-Kitaev systems exhibiting a Hawking-Page like phase transition. By computing the out-of-time-ordered correlator in the large $N$ limit by using the bilocal field formalism, we found that the chaos exponent of this model shows a discontinuous fall-off at the phase transition temperature. Hence in this model the Hawking-Page like transition is correlated with a transition in chaoticity, as expected from the relation between a black hole geometry and the chaotic behavior in the dual field theory.

## 1 Introduction

The Sachdev-Ye-Kitaev (SYK) model [2–4], a 1d system of $N$ Majorana fermions with all-to-all disordered interaction, is attracting attention for the following features. First, this model is considered to be dual to $AdS_2$ black hole geometry since they shares common descriptions at low energy (near the AdS boundary) [5]. The second interesting feature of the SYK model is that it is a highly tractable model with strongly chaotic dynamics.

Among the others there are the following two ways to characterize the chaoticity of a quantum system. One way is to use the fluctuation property of the energy levels: a quantum chaotic system shows the same level correlations as that of the random matrix ensemble which reflects the time reversal property of the system Hamiltonian [6], while for

a non-chaotic system the level fluctuations are not correlated to each other [7]. Another way is to use the out-of-time-ordered correlator (OTOC) [8]: for a quantum chaotic system the following four point function shows an exponential growth at late time for a generic choice of two operators $\widehat{V}, \widehat{W}$ [9]:

$$\text{Tr}\left[\widehat{V}\left(\frac{3\beta}{4} + it\right)\widehat{W}\left(\frac{\beta}{2}\right)\widehat{V}\left(\frac{\beta}{4} + it\right)\widehat{W}(0)e^{-\beta\widehat{H}}\right] \sim 1 - \frac{1}{L}e^{\lambda_L t}, \tag{1}$$

($L$: system size) where $\beta$ is the inverse temperature, which is a quantum analog of the initial value sensitivity $\Delta x(t)/\Delta x(0) \sim e^{\lambda_L^{(\text{cl})} t}$ in a classical chaotic system. In the SYK model the regularized OTOC (1) can be analyzed in the large $N$ limit [10]. In particular, in the strong coupling or the low temperature limit one can show analytically that the chaos exponent $\lambda_L$ saturates the bound [9]

$$\lambda_L \leq \frac{2\pi}{\beta}. \tag{2}$$

One can also study the level statistics by the numerical exact diagonalization of the Hamiltonian for each disorder realization, which indeed was found to coincides with that of the random matrix ensemble GUE/GOE/GSE depending on $N$ mod 8 [11–13].

On the other hand some quantum chaotic property can be derived holographically from the black hole geometry. For example the OTOC can be computed holographically as a scattering process near the horizon of the black hole [14]. This also supports the proposed duality between the SYK model and the $\text{AdS}_2$ black hole.

As the black hole geometry explains the dual field theory to be strongly chaotic, if we consider a model which exhibis a phase transition between a phase dual to black hole geometry and another phase which is not, it is expected that the chaotic property of the system also changes drastically at the phase transition [15]. This phase transition is known in the gravity side as the Hawking-Page transition [16] and was also realized in the field theory side in such as the four dimensional $\mathcal{N} = 4$ Yang-Mills theory [17]. Our aim is to realize the Hawking-Page like transition in a deformation of the SYK model, study its quantum chaotic property in detail and confirm whether the phase transition is accompanied with a transition in the chaoticity as expected.

Such model was already proposed in [18]. They considered the one dimensional quantum mechanical system with the following Hamiltonian

$$H = H_{\text{SYK}}(J_{ijk\ell}, \psi_i^L) + H_{\text{SYK}}(J_{ijk\ell}, \psi_i^R) + i\mu\sum_{i=1}^{\frac{N}{2}}\psi_i^L\psi_i^R, \tag{3}$$

where $H_{\text{SYK}}$ is the SYK Hamiltonian

$$H_{\text{SYK}}(J_{ijk\ell}, \psi_i) = \sum_{i<j<k<\ell}^{N} J_{ijk\ell}\psi_i\psi_j\psi_k\psi_\ell, \tag{4}$$

$\psi_i^L$ and $\psi_i^R$ are Majorana Fermions $\{\psi_i^a, \psi_j^b\} = \delta_{ab}\delta_{ij}$ and $J_{ijk\ell}$ are $\binom{N}{4}$ independent random variables obeying the following Gaussian distribution:

$$P(J_{ijk\ell}) = \sqrt{\frac{N^3}{12\pi\mathcal{J}^2}} \cdot \exp\left[-\frac{N^3}{24\mathcal{J}^2}J_{ijk\ell}^2\right], \qquad \text{(no sum)} \tag{5}$$

with $\mathcal{J} = 1$. Note that we have chosen the random couplings for L system and those for the R system perfectly correlated $J_{ijk\ell}^L = J_{ijk\ell}^R = J_{ijk\ell}$. From the analysis in the large

$N$ limit the following features were found. At low temperature the system is gapped due to the LR interaction term, and the system does not show the large $\mathcal{O}(N)$ entropy. This region corresponds to the global $\mathrm{AdS}_2$ spacetime where the two boundaries corresponds to the L/R SYK system. As the temperature is increased a new solution starts to exist which has the entropy of $\mathcal{O}(N)$ and corresponds to the two sided $\mathrm{AdS}_2$ black hole. In the canonical ensemble the system shows a phase transition between the two solutions. It was also found that when the LR coupling is larger than some critical value $\mu_c$ the two phases are smoothly connected and there are no phase transiion.

In [19] we studied the chaotic property of this model by using the level statistics. We found that the adjacent gap ratio [20] reproduces the value for the random matrix ensemble (GOE for this model) $r_{\mathrm{GOE}}$ for the bulk of the spectrum, while it takes substantially small value close to the edge of the spectrum. We also found that for $\mu \gtrsim \mu_c$ the adjacent gap ratio coincides with $r_{\mathrm{GOE}}$ for all region of the spectrum.

The observations in [19] suggests that the chaoticity of the two coupled model (3) depends on the energy scale and the transition may be correlated with the Hawking-Page like phase transition. However, since the analysis of the level statistics was restricted to finite $N$, it was not clear whether it was reasonable to compare our result with the large $N$ phase transition. To avoid this subtlety, in [1] we adopted the OTOC to characterize the quantum chaos which we can analyze directly in the large $N$ limit. As a result we found that the chaos exponent varies discontinuously at the phase trnasition temperature from the high temperature value $\lambda_L \sim \mathcal{O}(2\pi/\beta)$ to the low temperature value which is exponentially small with respect to the temperature.

This review is organized as follows. In section 2 we review the bilocal field ($G\Sigma$) formalism for the two coupled model in the large $N$ limit, where we also display the phase diagram which was obtained through this formalism. In section 3 we review the analytic continuation of the $G\Sigma$ formalism to the Lorentzian real time, briefly review the computation of the OTOC and display the results of the chaos exponent. In section 4 we summarize.

## 2 Bilocal field ($G\Sigma$) formalism and phase diagram

In this review we define the expectation value of an operator $\mathcal{O}[\psi_i^a]$ for the disordered system (3) as the disorder average of unnormalized vev divided by the disorder average of the partition function (annealed averaging) instead of the disorder average of the normalized vev (quenched averaging). For the quantities we consider in this review (the two point functions and the four point functions), one can show that the two results coincide in the large $N$ limit up to $\mathcal{O}(N^{-2})$. In this rule, the free energy at temperature $T = \beta^{-1}$ is given as

$$F(\beta) = -\frac{1}{\beta} \log\langle Z(\beta)\rangle_{J_\alpha}, \tag{6}$$

where $Z(\beta)$ is the thermal partition function

$$\langle Z(\beta)\rangle_{J_\alpha} = \int \prod_{i<j<k<\ell} dJ_{ijk\ell} P(J_{ijk\ell}) \int \prod_{a,i} \mathcal{D}\psi_i^a(\tau) e^{-\int_0^\beta d\tau(\sum_{i,a}\psi_i^a \partial_\tau \psi_i^a + H)}, \tag{7}$$

with $P(J_{ijk\ell})$ and $H$ defined in (5) and (3).

The partition function (7) can be rewritten by using the bilocal field $G_{ab}(\tau_1, \tau_2) =$

$(1/N) \sum_{i=1}^{N} \psi_i^a(\tau_1)\psi_i^b(\tau_2)$ as [1, 18]

$$\langle Z(\beta)\rangle_{J_\alpha} = \int \prod_{a,b} \mathcal{D}G_{ab}(\tau_1,\tau_2)\mathcal{D}\Sigma_{ab}(\tau_1,\tau_2)e^{-NS_{\text{eff}}[G_{ab},\Sigma_{ab}]}, \tag{8}$$

with

$$S_{\text{eff}}[G_{ab},\Sigma_{ab}]$$

$$= -\frac{1}{4}\log\det\begin{pmatrix} -\delta(\tau-\tau')\partial_{\tau'} + \frac{\Sigma_{LL}(\tau,\tau')-\Sigma_{LL}(\tau',\tau)}{2} & \frac{\Sigma_{LR}(\tau,\tau')-\Sigma_{RL}(\tau',\tau)}{2} - i\mu\delta(\tau-\tau') \\ \frac{\Sigma_{RL}(\tau,\tau')-\Sigma_{LR}(\tau',\tau)}{2} + i\mu\delta(\tau-\tau') & -\delta(\tau-\tau')\partial_{\tau'} + \frac{\Sigma_{RR}(\tau,\tau')-\Sigma_{RR}(\tau',\tau)}{2} \end{pmatrix}$$

$$+ \sum_{a,b}\frac{1}{4}\int d\tau d\tau'\Big(\Sigma_{ab}(\tau,\tau')G_{ab}(\tau,\tau') - \frac{\mathcal{J}^2}{2}G_{ab}(\tau,\tau')^4\Big). \tag{9}$$

Here $\Sigma_{ab}(\tau_1,\tau_2)$ are auxiliary fields introduced to treat $G_{ab}(\tau_1,\tau_2)$ as independent integration variables from $\psi_i^a(\tau)$.

The overall factor $N$ in the exponent (8) implies that in the large $N$ limit the partition function can be evaluated by the saddle point approximation

$$\langle Z(\beta)\rangle_{J_\alpha} \approx \sum_{\text{saddles}} e^{-NS_{\text{eff}}[G_{ab}^{(\text{saddle})},\Sigma_{ab}^{(\text{saddle})}]}, \tag{10}$$

where $G_{ab}^{(\text{saddle})}, \Sigma_{ab}^{(\text{saddle})}$ are the solutions to the equations of motion $\delta S_{\text{eff}}/\delta G_{ab} = \delta S_{\text{eff}}/\delta \Sigma_{ab} = 0$, or explicltly [1]

$$\partial_{\tau_1}G_{ab}(\tau_1,\tau_2) - \sum_c\Big(-i\mu\epsilon_{ac}G_{cb}(\tau_1,\tau_2) + \int_0^\beta d\tau_3\Sigma_{ab}(\tau_1,\tau_3)G_{cb}(\tau_3,\tau_2)\Big) = \delta_{ab}\delta(\tau_1-\tau_2),$$

$$\Sigma_{ab}(\tau_1,\tau_2) = 2\mathcal{J}^2 G_{ab}(\tau_1,\tau_2)^3. \tag{11}$$

Solving the equations of motion (11) numerically, we obtain two different solutions; one exists for $T > t_{c,\text{BH}}$ and the other exists for $T < T_{c,\text{WH}}$, with some $\mu$-dependent temperatures $T_{c,\text{BH}},T_{c,\text{WH}}$ ($T_{c,\text{BH}} < T_{c,\text{WH}}$), which corresponds respectively to the two sided AdS$_2$ black hole and the global AdS$_2$ (eternal traversable wormhole) [18]. In figure 1 (left) we have displayed $S_{\text{eff}}$ evaluate at each solution. For $T_{c,\text{BH}} < T < T_{c,\text{WH}}$ both of the two solutions exist and the values of $S_{\text{eff}}$ intersect at some $T_c$, where the system exhibits a first order phase transition. The coexisting region $T_{c,\text{BH}} < T < T_{c,\text{WH}}$ becomes narrower as $\mu$ increases, and it disappears at $\mu = \mu_c \approx 0.177$ (figure 1 (right)). For $\mu > \mu_c$ there is only one solution which is continuously connected to both of the black hole solution and the wormhole solution, and there are no phase transition.

## 3  Out-of-time ordered correlator and chaos exponent

We consider the following OTOC

$$\frac{1}{N^2}\sum_{i,j=1}^{N}\langle \psi_i^a(u_1)\psi_i^b(u_2)\psi_j^c(u_3)\psi_j^d(u_4)\rangle, \tag{12}$$

with $u_1 = 3\beta/4 + it_1$, $u_2 = \beta/4 + it_2$, $u_3 = \beta/2$, $u_4 = 0$. This quantity is written in the $G\Sigma$ formalism as

$$\int\Big(\prod_{a,b}\mathcal{D}G_{ab}\mathcal{D}\Sigma_{ab}\Big)G_{ab}(u_1,u_2)G_{cd}(u_3,u_4)e^{-NS_{\text{eff}}[G_{ab},\Sigma_{ab}]}. \tag{13}$$

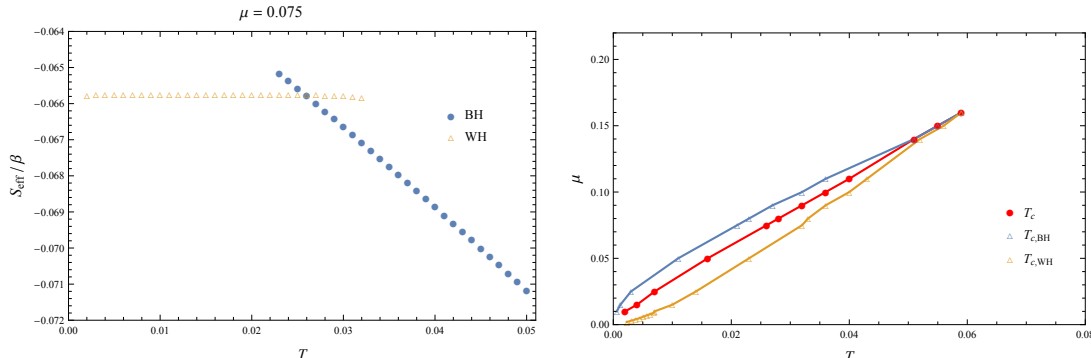

Figure 1: Left: $S_{\text{eff}}$ evaluated with the two solutions of the equations of motion (11); Right: The temperatures $T_{c,\text{WH}}, T_{c,\text{BH}}$ at which one of the two solutions disappears and the critical temperature $T_c$.

If $u_1, u_2, u_3, u_4$ were real numbers, the OTOC (12) could be evaluated in the large $N$ limit by expanding $G_{ab}, \Sigma_{ab}$ around the dominant saddle, as [1]

$$\frac{1}{N^2} \sum_{i,j=1}^{N} \langle \psi_i^a(\tau_1) \psi_i^b(\tau_2) \psi_j^c(\tau_3) \psi_j^d(\tau_4) \rangle$$

$$= G_{ab}^{(0)}(\tau_1, \tau_2) G_{cd}^{(0)}(\tau_3, \tau_4) + \frac{1}{N} \mathcal{F}_{abcd}(\tau_1, \tau_2, \tau_3, \tau_4) + \mathcal{O}(N^{-2}), \tag{14}$$

with

$$\mathcal{F}_{abcd}(\tau_1, \tau_2, \tau_3, \tau_4)$$

$$= \mathcal{F}_{0,abcd}(\tau_1, \tau_2, \tau_3, \tau_4) + \sum_{e,f} \int_0^\beta d\tau d\tau' \mathcal{K}_{abef}(\tau_1, \tau_2, \tau, \tau') \mathcal{F}_{efcd}(\tau, \tau', \tau_3, \tau_4),$$

$$\mathcal{F}_{0,abcd}(\tau_1, \tau_2, \tau_3, \tau_4) = -G_{ac}^{(0)}(\tau_1, \tau_3) G_{bd}^{(0)}(\tau_2, \tau_4) + G_{ad}^{(0)}(\tau_1, \tau_4) G_{bc}^{(0)}(\tau_2, \tau_3),$$

$$\mathcal{K}_{abcd}(\tau_1, \tau_2, \tau_3, \tau_4) = -6\mathcal{J}^2 G_{ac}^{(0)}(\tau_1, \tau_3) G_{bd}^{(0)}(\tau_2, \tau_4) G_{cd}^{(0)}(\tau_3, \tau_4)^2. \tag{15}$$

The actual OTOC (12) can be obtained by analytically continuing these results. In the path integral formalism of a quantum mechanical problem, the time evolution of an inserted operator in the operator formalism $\langle \psi | \cdots e^{i\widehat{H}t} \widehat{\mathcal{O}} e^{-i\widehat{H}t} \cdots | \psi \rangle$ results in a non-single valued configuration of the path integral fields (see figure 2 (left)). This effect can be taken care of by introducing two different components $>, <$ of the bilocal fields

$$G_{ab}^{>}(t_1, t_2) = -i \lim_{\epsilon \to +0} G_{ab}(\epsilon + it_1, -\epsilon + it_2),$$

$$G_{ab}^{<}(t_1, t_2) = -i \lim_{\epsilon \to +0} G_{ab}(-\epsilon + it_1, \epsilon + it_2), \tag{16}$$

and the equations of motion (11) are continued as [1]

$$\begin{pmatrix} \widetilde{G}_{LL}^R(\omega) & \widetilde{G}_{LR}^R(\omega) \\ \widetilde{G}_{RL}^R(\omega) & \widetilde{G}_{RR}^R(\omega) \end{pmatrix} = \frac{1}{(\omega - \widetilde{\Sigma}_{LL}^R(\omega))(\omega - \widetilde{\Sigma}_{RR}^R(\omega)) - (\widetilde{\Sigma}_{LR}^R(\omega) + i\mu)(\widetilde{\Sigma}_{RL}^R(\omega) - i\mu)}$$

$$\times \begin{pmatrix} \omega - \widetilde{\Sigma}_{RR}^R(\omega) & \widetilde{\Sigma}_{LR}^R(\omega) + i\mu \\ \widetilde{\Sigma}_{RL}^R(\omega) - i\mu & \omega - \widetilde{\Sigma}_{LL}^R(\omega) \end{pmatrix},$$

$$G_{ab}^R(t) = \int_{-\infty}^{\infty} \frac{d\omega}{2\pi} e^{-i\omega t} \widetilde{G}_{ab}^R(\omega), \quad \Sigma_{ab}^{>}(t) = -\frac{\mathcal{J}^2}{4} G_{ab}^{>}(t)^3, \quad \Sigma_{ab}^{<}(t) = -\frac{\mathcal{J}^2}{4} G_{ab}^{<}(t)^3, \tag{17}$$

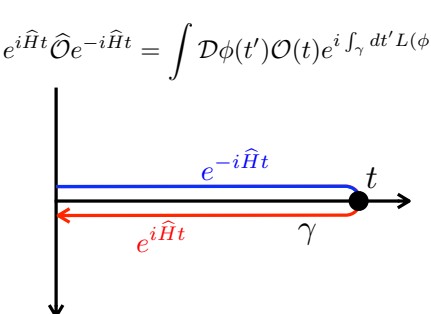

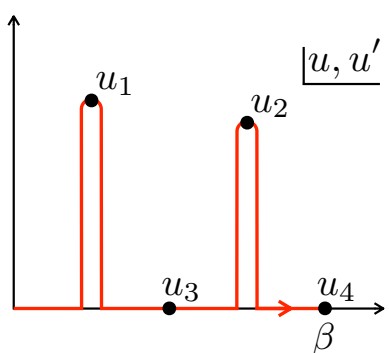

Figure 2: Left: Path integral contour corresponding to the real time evolution of an operator for a generic quantum mechanical problem. The configuration of $\phi(t)$ on the blue line and that on the red line are regarded as independent path integral variables. Right: The integration contour $C$ for the real time continuation of the ladder equation (15).

where

$$G_{ab}^R(t) = \theta(t)(G_{ab}^>(t) - G_{ab}^<(t)). \tag{18}$$

Note that in (17) and (18) we have assumed that the configurations of the bilocal fields $G_{ab}(u_1, u_2), \Sigma_{ab}(u_1, u_2)$ depends on $u_1, u_2$ only through the difference $G_{ab}(u_1, u_2) = G_{ab}(u_1 - u_2), \Sigma_{ab}(u_1, u_2) = \Sigma_{ab}(u_1 - u_2)$. The equations of motion (17) gives a closed system together with the relation between $>$ component and the retarded component (18), and the following KMS condition with temperature $T = \beta^{-1}$:

$$G_{ab}(u) = i \int_{-\infty}^{\infty} \frac{d\omega}{2\pi} e^{-\omega u} \frac{\widetilde{G}_{ab}^R(\omega) - (\widetilde{G}_{ab}^R(\omega))^*}{1 + e^{-\beta\omega}}. \qquad (u = \tau + it, \quad 0 < \tau < \beta) \tag{19}$$

The ladder equation for real time OTOC is obtained from (15) by replacing $G_{ab}^{(0)}$ with the solution of the real time equations of motion (17)-(19) and the integration contour $\int_0^\beta d\tau d\tau'$ according to the rule depicted in figure 2 (left), which result in $\int_C du du'$ with the contour $C$ depicted in figure 2 (right). If we further assume that $\mathcal{F}_{abcd}(u_1, u_2, u_3, u_4)$ grows exponentially as $\mathcal{F}_{abcd}(u_1, u_2, u_3, u_4) \approx e^{\lambda_L(t_1+t_2)/2} f_{abcd}(t_{12})$ and keep only the contributions relevant to this late time growth, we end up with

$$f_{abcd}(t_{12}) \approx -6\mathcal{J}^2 \sum_{e,f} \int_{-\infty}^{\infty} dt e^{-\frac{\lambda_L(t_{12}-t)}{2}} \left[ \int_{-\infty}^{\infty} dt' G_{ae}^{(0)R}(t_{12} - t - t') G_{bf}^{(0)R}(-t') e^{\lambda_L t'} \right]$$

$$G_{ef}^{(0)}\left(\frac{\beta}{2} + it\right) f_{efcd}(t). \tag{20}$$

The real time ladder eqaution (20) has a non-trivial solution $f_{abcd}(t)$ only if $\lambda_L$ is not larger than the actual value of the chaos exponent (1). Hence we can obtain the chaos exponent by varying $\lambda_L$ and finding the value where the largest eigenvalue of the ladder operation in the right-hand side of (20) crosses 1. This procedure can be performed numerically and we obtain the chaos exponent as figure 3.

# 4  Conclusion

In this article we have briefly reviwed [1] where we have computed the chaos exponent of the model (3) consisting of two SYK systems coupled by a uniform quadratic interaction.

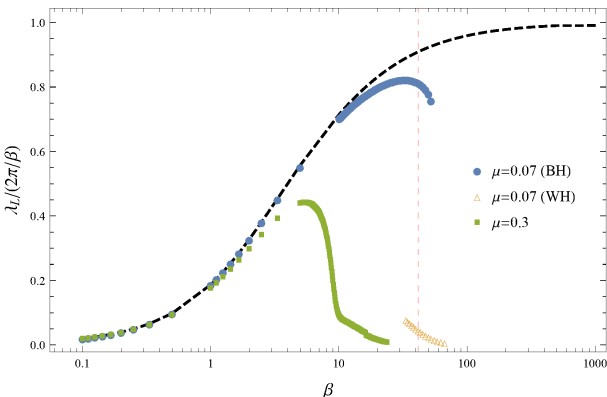

Figure 3: Chaos exponent compared with the value for pure SYK (dashed black line). The vertical dashed red line is $T = T_c(\mu = 0.07) \approx 0.024$.

This two coupled model exhibits a Hawking-Page like phase transition in the large $N$ limit for $\mu < \mu_c \approx 0.177$ [18], as displayed in figure 1. In [1] we computed the chaos exponent of this model in the large $N$ limit in the whole parameter regime including the region close to the phase transition point. As a result we found that as the temperature is decreased the chaos exponent varies discontinuously at the phase transition point $T = T_c$ from the value of order the chaos bound $2\pi/\beta$ to an extremely small value, as displayed in figure 3. This result is in agreement with our expectation that the Hawking-Page like transition would be correlated with a transition in the chaoticity [15, 19].

In the large $N$ limit both the free energy and the out-of-time-ordered correlators are determined by the bilocal fields $G_{ab}(u_1, u_2)$ satisfying the equations of motion (11) and (17)-(19). Although the Euclidean equations of motion (11) and its real time continuation (17)-(19) are slightly different, the solutions are always in one-to-one correspondence (19). Therefore, at $T_c$, where the Euclidean solution giving dominant contribution to the free energy changes from the black hole solution to the wormhole solution, the real time solution also switches, hence the chaos exponent changes discontinuously from the value for the black hole solution to the value for the wormhole solution. From this viewpoint the correlation between the Hawking-Page like phase transition and the transition in the chaoticity for an SYK-like model would be trivial. Nevertheless, it was still non-trivial how the chaos exponent for the two solutions behaves.

Another interesting point in our result is that the chaos exponent is small but non-zero even in the wormhole phase; the system is weakly chaotic even in the wormhole phase. This is surprising but consistent with the fact that the two point function decays exponentially even in the wormhole phase [21], which is another criterion for the quantum chaos. Note that this is not a generic feature of the systems exhibiting the Hawking-Page like phase transition. For example in the four dimensional $\mathcal{N} = 4$ Yang-Mills theory on $S^3$ in the weak coupling limit [17], a two point function does not show an exponential decay in the low temperature confined phase [22, 23]. It would be interesting to clarify what the origin for this difference is.

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
