# Peer review of "A sharp transition in quantum chaos and thermodynamics of mass deformed SYK model"

_SciPost Physics Proceedings_

## Round 1 · Referee Report · Anonymous (Referee 1) · 2020-11-16

Report

The author presented a nice review of his research work published earlier. The motivation and the set-up of the research were stated concisely. Given the limited space, the author presented the result of his work with the aid of plots in an effective way.

As this is a review article, one suggestion which might benefit the readers is to focus on the explanation of physics a bit more. For example, the discussion (page 4) about the phase diagram with respect to the temperature and the coupling ($\mu$) is very interesting but somewhat limited. A slightly more elaborated discussion or explanation about the physics or implication behind the result would definitely be welcome.

The review is interesting and well written.

Requested changes

1- Just a small typo. On page 2, 9th line below equation (2), it should read "exhibits" instead of "exhibis".

2- Please state the definition of $\left\langle \dots \right\rangle_{ J_\alpha}$ in equation 7. (I presume it refers to disorder average discussed in the first paragraph of section 2 but a clear definition would be helpful.)

3-Similarly for equation 12, it is helpful to define the meaning of expectation value. (Thermal average etc.)

---

## Editorial Decision

resubmitted